# Towards Multimodal Open Set Recognition

## Abstract

Open set recognition (OSR) requires deep learning models to identify unknown samples while recognizing known ones. Existing OSR studies focus on single-modal data but merely discuss how to handle multimodal data. In this paper, we propose a new task multimodal open set recognition (MMOSR), extending OSR to more practical scenarios. First, we analyze the necessity of MMOSR and provide insights into the task. We find that simply combining OSR and multimodal fusion methods faces the challenge of fusion degradation. The main reason is that the OSR regularization constrains the fused representations to be excessively compact, leading to deactivated and limited representations. We design the multimodal representation reactivation network (MRN) to alleviate fusion degradation by reactivating suppressed representations. MRN includes the mutually enhanced fusion for enhancing representations and performing cross-modal interaction, and the adaptive fusion for capturing multiple informative representations and outputting the adaptively fused prediction. Thus, the proposed method obtains effective and comprehensive multimodal representations and addresses the challenge of fusion degradation. Finally, extensive experiments on various settings demonstrate that the proposed method is superior to existing methods by up to 5.23% on OSCR.

## 1 Introduction

Deep neural networks can accurately identify samples from learned classes during training but struggle to recognize samples from unknown classes (Geng et al., 2020). Open set recognition (OSR) is proposed to enable models to classify known samples while rejecting unknown ones, which is the fundamental ability of models to ensure security and know what to learn (Scheirer et al., 2012).

Many researchers have dedicated to the OSR problem on various types of data, such as images, texts, and time series (Huang et al., 2023; Liu et al., 2023; Yang et al., 2022). Existing OSR methods can be roughly divided into two groups: generative methods that encode classes into various latent distributions or synthesizing pseudo-unknown samples for constraining the space occupied by known classes (Katsumata et al., 2022; Kong & Ramanan, 2021; Yue et al., 2021) and discriminative methods that reserve space for unknown classes by incorporating placeholders or reciprocal points (Chen et al., 2020; 2022; Zhou et al., 2021; Xu et al., 2023).

Despite the progress in OSR, existing studies work upon single-modal data but merely address multimodal data. In fact, humanoid robots (Hirose & Ogawa, 2007) and unmanned systems typically rely on multimodal sensors, *e.g.*, image-text or vision-audio data, to perceive the environment, which challenges the mechanism of single-modal OSR methods that requires comprehensively capturing unknown factors. To address this limitation, we propose a new multimodal open set recognition (MMOSR) task, analyzing its necessity and challenge by discussing the following questions:

*(1) Is it possible to solve MMOSR using existing multimodal fusion and single-modal OSR methods?*

*(2) If not, what is the main challenge of applying these existing methods to MMOSR?*

*(3) How to address the aforementioned challenge and design effective MMOSR methods?*

To answer these questions, we empirically analyze three types of models: single-modal OSR, multimodal fusion, and the combination of multimodal fusion and single-modal OSR. As illustrated in Figure 1, we find that combining multimodal learning and single-modal OSR cannot perform satisfactorily (Section 3.2). OSR regularization leads to the collapse of multimodal representations by suppressing the fused representations of each class to be excessively compact. Such an excessively

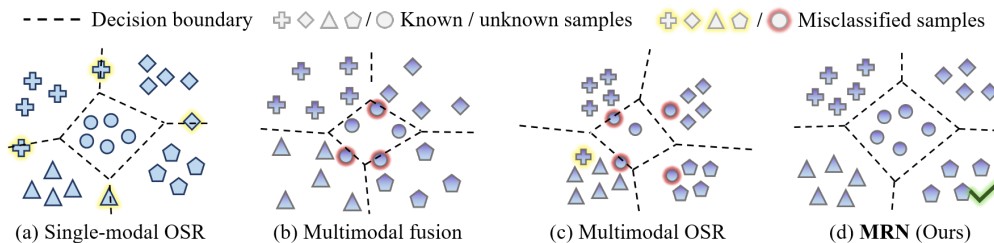

Figure 1: Illustration of MMOSR issues. (a) Single-modal OSR accurately detects unknowns but misclassifies some known samples, (b) Multimodal fusion improves accuracy but struggles with unknowns, (c) Multimodal OSR over-suppresses the feature space and hinders the performance, (d) **MRN** extracts discriminative features comprehensively, enhancing overall performance.

compact suppression results in fusion degradation, which specifically manifests as a suppression of modality representations and limited representation capability. Fusion degradation reduces the information learned in a certain modality and makes the model focus only on partial information. Therefore, the performance significantly decreases with weakened multimodal representations.

To address the aforementioned challenge, we propose the multimodal representation reactivation network (MRN), which reactivates representations and learns comprehensive representations through effective fusion across modalities. Concretely, to alleviate fusion degradation, we design an adaptive fusion module using a mixture of experts to obtain comprehensive representations. To avoid the fusion suppression of modality, we design a mutually enhanced fusion module to enable full interaction between different modalities. With effective fusion modules in the model, the learned representations are more comprehensive and informative. Extensive experiments show that our method achieves an over 5.23% improvement on OSCR over existing methods, thereby verifying its effectiveness.

The contributions of the paper are three-fold:

- We propose a new multimodal open set recognition (MMOSR) task that extends the existing OSR task to handling multimodal data.
- We provide insights into MMOSR and find that simply combining single-modal OSR and multimodal fusion would lead to the challenge of fusion degradation.
- We design a multimodal representation reactivation network with mutually enhanced fusion and adaptive fusion to enable comprehensive and informative representations.

## 2 RELATED WORK

### 2.1 MULTIMODAL FUSION

Multimodal fusion aims to integrate information from diverse sensory modalities (such as vision, audio, and text) to achieve a more comprehensive and accurate understanding. Simple operations like addition and concatenation enable the integration of representations from diverse sources (Nojavanasghari et al., 2016; Wang et al., 2017). TMC (Han et al., 2023) leverages variational Dirichlet distribution and Dempster-Shafer theory to improve accuracy and robustness by dynamically integrating views based on uncertainty. GQA (Ainslie et al., 2023) improves upon multi-query attention by grouping query heads, achieving near multi-head attention quality with faster inference speed through minimal uptraining. MLA (Zhang et al., 2024) improves multimodal learning by alternating unimodal adaptations to reduce interference between modalities and optimize cross-modal interactions.

Moreover, large-scale pre-trained models excel in image-text classification. CLIP (Radford et al., 2021) employs contrastive learning to align visual and textual representations in a shared embedding space, CoOp (Zhou et al., 2021) extends CLIP by optimizing learnable prompts to adapt to new downstream tasks, while MaPLe (Khattak et al., 2023) further enhances this by introducing multiple prompt vectors that enrich prompt diversity and enhance the connections between image and text.

Even with the recent developments in multimodal fusion (MM) methods, they neglect unknown samples that may appear, leading to known classes dominating feature space and hindering the detection of unknown samples. Intuitively, we investigate combining MM and OSR to handle the MMOSR task in Section 3, but the performance remains unsatisfactory.

## 2.2 OPEN SET RECOGNITION

Early OSR methods used conventional machine learning techniques like support vector machines and extreme value theory (Bendale & Boult, 2016; Scheirer et al., 2012). The maximum Softmax probability served as the baseline for rejecting unknown samples, representing the highest probability of a sample belonging to a known class (Hendrycks & Gimpel, 2016).

Recent OSR methods based on deep learning can be categorized into generative and discriminative methods. Generative methods aim to generate pseudo-unknown samples, enabling models to handle and distinguish unknown samples accurately (Katsumata et al., 2022; Kong & Ramanan, 2021; Yue et al., 2021). Several generative methods incorporate an additional component that learns to reconstruct inputs from features, using the reconstruction errors to estimate whether the test sample is from known classes (Huang et al., 2023; Oza & Patel, 2019; Perera et al., 2020; Yoshihashi et al., 2019). Discriminative methods use specific classification strategies for handling OSR tasks, such as using prototype points as representatives for known classes or setting placeholders to reserve space for unknown samples (Chen et al., 2020; 2022; Xu et al., 2023; Zhou et al., 2021).

Despite recent advances, existing OSR methods are mainly for single-modal tasks. Extending OSR to MMOSR using multimodal fusion models before applying OSR strategies is straightforward. However, exploratory experiments show that simply combining MM and OSR does not achieve satisfactory performance, highlighting the unique challenge in MMOSR.

## 3 MMOSR

In this section, we conducted experiments on the proposed MMOSR benchmark to illustrate the challenge of handling the MMOSR task and the necessity of developing specific MMOSR methods.

### 3.1 PROBLEM DEFINITION

In multimodal open set recognition (MMOSR) task, the $i$-th sample $(\mathbf{x}_i^1, ..., \mathbf{x}_i^{|\mathcal{M}|}, y_i)$ comprises data from $|\mathcal{M}|$ modalities and the corresponding label. The training set $\mathcal{D}_K = \{(\mathbf{x}_i^1, ..., \mathbf{x}_i^{|\mathcal{M}|}, y_i)\}_{i=1}^n$ only contains samples from known classes, whereby $y_i \in \{1, ..., C_K\}$ and $C_K$ is the number of known classes. The traditional multimodal classification task aims to classify multimodal data under closed-set assumption. However, under the MMOSR setting, the test set $\mathcal{D}_T = \{(\mathbf{x}_j^1, ..., \mathbf{x}_j^{|\mathcal{M}|}, y_j)\}_{j=1}^t$ includes samples from both known classes and unknown classes, where the label $y_j \in \{1, ..., C_K\} \cup \{C_K + 1, ..., C_K + C_U\}$ and $C_U$ represents the number of unknown classes that appear in testing scenario. Since we cannot obtain any prior information about real scenarios, the model should assign all unknown samples to the unified unknown class as $U$.

### 3.2 NECESSITY OF DEVELOPING MMOSR METHODS

With the MMOSR problem defined, we now discuss why existing methods are inadequate and explore the necessity of developing specialized MMOSR approaches.

We first conducted experiments on the widely used image-text multimodal dataset Food-101 (Wang et al., 2015), which includes food images paired with recipe texts across 101 categories. Following the OSR setting (Chen et al., 2022), we randomly selected 5, 10, or 20 classes from Food-101 as known classes for testing, while samples from the remaining classes were treated as unknown during testing. We applied the recent single-modal OSR method OpenAUC (Wang et al., 2022) to either the image or text modality (Image/Text-OSR).

Table 1: Comparison of AUROC and ACC on different methods under MMOSR settings. The best results are marked in **bold**.

| Method | Food-101 / 5 | | Food-101 / 10 | | Food-101 / 20 | |
|---|---|---|---|---|---|---|
| | AUROC | ACC | AUROC | ACC | AUROC | ACC |
| Image-OSR | 65.91 | 77.57 | 62.21 | 60.24 | 60.07 | 33.37 |
| Text-OSR | **91.57** | 91.17 | **90.49** | 89.24 | **92.03** | 87.45 |
| Fusion | 89.34 | **93.24** | 90.48 | **92.24** | 91.33 | **91.37** |
| *Gain* ($\Delta$) | (2.23↓) | (0.33↑) | (0.01↓) | (3.00↑) | (0.70↓) | (3.88↑) |
| Fusion-OSR | 89.23 | 92.91 | 88.24 | 87.01 | 86.43 | 85.52 |
| *Gain* ($\Delta$) | (2.34↓) | (0.33↓) | (2.25↓) | (5.23↓) | (5.60↓) | (5.85↓) |

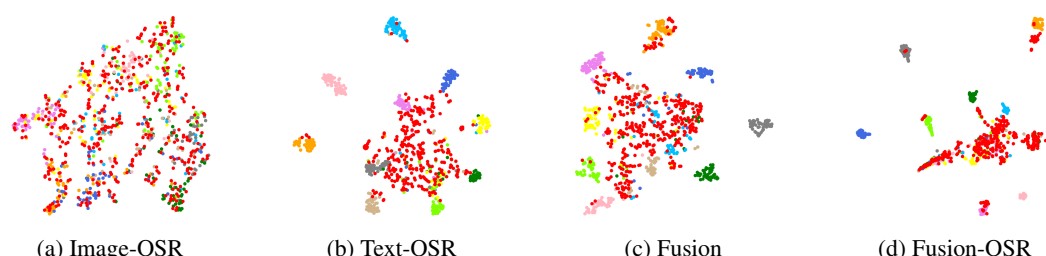

|  (a) Image-OSR | (b) Text-OSR | (c) Fusion | (d) Fusion-OSR |

Figure 2: Comparison of t-SNE downscaled features for each method under the 10 known classes setting on Food-101. The red dots represent downscaled representations of unknown samples, while each dot of the same color represents known samples of the same class.

For MM methods, we used a simple fusion approach by adding representations from both modalities and rejecting unknown samples based on the maximum Softmax probability strategy (Fusion). We reported average accuracy (ACC) and the area under the receiver operating characteristic curve (AUROC) in Table 1 to evaluate the model's closed-set classification performance and its ability to detect unknown samples. Two key conclusions can be inferred from the results:

**Inadequacy of MM and OSR for MMOSR tasks.** As depicted in Table 1, applying OSR methods solely to single-modality data results in low ACC, likely due to the limited information from a single modality. Though MM improves ACC performance, its performance of AUROC in detecting unknown samples remains lower than the most discriminative modality (Text).

**Limitations of simply combining MM and OSR methods.** Relying solely on MM fails to recognize unknown samples effectively, while using OSR on a single modality leads to significant information loss. Combining MM representations with OSR (Fusion-OSR) results in lower ACC and AUROC than either method alone. Our experiments reveal that MM methods lack measures to mitigate open space risk, which cannot be resolved by simply integrating OSR methods. We propose that MMOSR should leverage the strengths of both MM and OSR to enhance closed-set and open-set performance.

To analyze the key challenge in the MMOSR task, Figure 2 visualizes the downscaled representations of each model. Based on earlier experiments, we identified a critical issue termed **fusion degradation**, where basic fusion approaches reduce the distinctiveness of each modality's representation. We explain this phenomenon from the following two perspectives:

**Weakened discrimination ability of the model on fused representations.** Comparing Figure 2a, Figure 2b, and Figure 2d, we observe that due to high similarity among images from different known classes, leveraging the text modality through fusion is crucial. However, when OSR methods use fused representations (Fusion-OSR), the distinction between unknown and known samples diminishes, as shown by the close alignment of unknown samples with known class clusters. Results in Table 1 also suggest that fusion suppresses single-modal representations and reduces the model's discrimination ability. This phenomenon indicates that existing MM methods are unsuitable for OSR tasks.

**Over-compression of representations in OSR methods.** To handle the risk of misclassifying unknown samples, OSR methods compact the representations of known classes to leave space for unknown samples. However, as seen in Figure 2c and Figure 2d, Fusion-OSR methods over-compress these representations. This excessive compression limits the model's ability to represent and differentiate unknown classes, causing unknown samples to closely resemble known clusters.

## 4 METHOD

In this section, we present our multimodal representation reactivation network (MRN) for MMOSR, which can obtain more effective and comprehensive representations that are fused from multiple modalities to classify known samples while rejecting unknown ones precisely.

### 4.1 FRAMEWORK

Figure 3 illustrates the architecture of the proposed MRN, which employs data encoders and fusion modules. The fusion modules consist of a mutually enhanced fusion module that reactivates significant

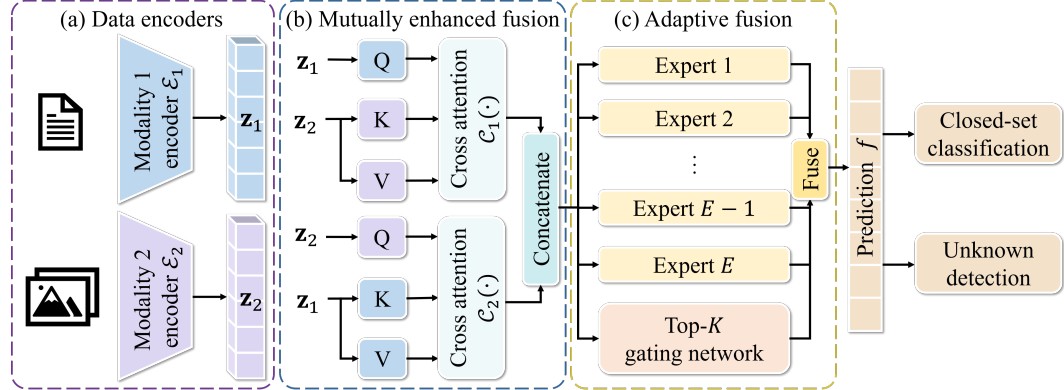

Figure 3: The architecture of the MRN includes three modules: (a) encoder $\mathcal{E}$ for each modality, (b) mutually enhanced fusion, and (c) adaptive fusion. $f$ represents the fused prediction used for closed-set classification and unknown detection.

representations across modalities and an adaptive fusion module that extracts and combines multiple representations from experts, ensuring a comprehensive and effective representation for classification.

## 4.2 MULTIMODAL REPRESENTATION REACTIVATION NETWORK

To explain our method clearly, we illustrate it using dual-modality data, where the training sample is represented as $(\mathbf{x}_1, \mathbf{x}_2, y)$. We first input modality 1 data $\mathbf{x}_1$ and modality 2 data $\mathbf{x}_2$ into the corresponding encoder for obtaining the representations separately: $\mathbf{z}_1 = \mathcal{E}_1(\mathbf{x}_1), \mathbf{z}_2 = \mathcal{E}_2(\mathbf{x}_2)$. Then, we input the representations $\mathbf{z}_1, \mathbf{z}_2$ into the following fusion modules to obtain effectively fused representations with deeply cross-modal interaction for prediction.

### 4.2.1 MUTUALLY ENHANCED FUSION OF REPRESENTATIONS

Experiments in Section 3.2 show that existing methods suffer from fusion degradation, affecting their ability to utilize comprehensive information to distinguish unknown samples from known ones. It is necessary to ensure that the fused representations contain key information from multiple modalities. In each modality, there is a sharing of class-irrelevant information across classes, which is not advantageous for the OSR task. Thus, guiding the model to focus on class-relevant significant information within each modality can enhance representations.

Inspired by the study of Katharopoulos et al. (2020) on attention mechanisms, we use cross-attention to enhance representations of each modality mutually. We consider that the parts where single modality representations highly correlate with representations from another modality represent the class-relevant information that needs to be retained. Therefore, we use cross-attention modules $\mathcal{C}_1$ and $\mathcal{C}_2$, each dedicated to extracting the crucial information from the corresponding modality, thereby mutually reactivating suppressed representations and interacting between different modalities.

For module $\mathcal{C}_1$, we use the modality 1 representations $\mathbf{z}_1$ as the query and the modality 2 representations $\mathbf{z}_2$ as the key and value. As in equation 1, the attention scores represent the relative importance of each element in $\mathbf{z}_2$ concerning the query $\mathbf{z}_1$. The normalized attention scores are then used to compute a weighted sum of $\mathbf{z}_2$ and output $\mathbf{z}_1^c$, which captures the most relevant information from $\mathbf{z}_2$.

$$\mathbf{z}_1^c = \mathcal{C}_1\left(\mathbf{z}_1, \mathbf{z}_2, \mathbf{z}_2\right) = \mathrm{Softmax}\left(\mathbf{W}_1^Q \mathbf{z}_1 \mathbf{z}_2 \mathbf{W}_1^K / \sqrt{d}\right)\left(\mathbf{W}_1^V \mathbf{z}_2\right), \tag{1}$$

where $d$ represents the dimension of the query vectors and $\mathbf{W}_1^*$ denotes the learnable parameter matrix designed to map the original vectors into a representation space that can better match other vectors, enabling the model to understand the relationship between representations.

Similarly, we use $\mathbf{z}_2$ as the query and $\mathbf{z}_1$ as the key and value for $\mathcal{C}_2$, and obtain $\mathbf{z}_2^c = \mathcal{C}_2(\mathbf{z}_2, \mathbf{z}_1, \mathbf{z}_1)$. We adopt multi-head attention for combining weights to yield richer and more accurate representations. The final output of the mutually enhanced fusion module is $\mathbf{z}_c = \mathrm{Concatenate}(\mathbf{z}_1^c, \mathbf{z}_2^c)$.

By interacting multimodal representations with cross-attention, the model mutually enhances each modality's representation, reactivating suppressed features. For data with three or more modalities, pairwise enhancement followed by concatenation can be applied.

### 4.2.2 ADAPTIVE FUSION OF REPRESENTATIONS

To address the issue of the limited representation capability of the model, we introduce Mixture-of-Experts (MoE) (Shazeer et al., 2016) to extract diverse representations and adaptively obtain the fused prediction. By leveraging specialized experts and adaptive weights, the model can reactivate the important but subtle representations then utilize comprehensive representations for prediction.

The adaptive fusion comprises two main components: the expert network $\mathcal{N}(\cdot)$ and the gating network $\mathcal{G}(\cdot)$. The expert network $\mathcal{N}(\cdot)$ consists of multiple multilayer perceptron (MLP) experts, each capable of extracting different aspects or patterns from the data. Let $E$ denotes the number of experts, and the expert network outputs a set of predictions $\mathbf{f}_a = \mathcal{N}(\mathbf{z}_c) = \{\mathbf{f}_a^e\}_{e=1}^E \in \mathbb{R}^{[E, C_K]}$. The gating network $\mathcal{G}(\cdot)$ determines the contribution of each expert to the final prediction based on the input $\mathbf{z}_c$. In equation 2, $\mathcal{G}(\cdot)$ computes expert weights $\mathbf{w}_a$ by applying Softmax to the top-$K$ elements of the dot product between $\mathbf{z}_c$ and parameter matrix $\mathbf{W}_g$, with Gaussian noise $\epsilon \sim \mathcal{N}(0, \mathbf{I})$ added for robustness and to prevent overfitting.

$$\mathbf{w}_a = \mathcal{G}(\mathbf{z}_c) = \mathrm{Softmax}\left(\mathrm{top-}K\left((\mathbf{W}_g \mathbf{z}_c) + \epsilon \cdot \mathrm{Softplus}\left(\mathbf{W}_{\mathrm{noise}} \mathbf{z}_c\right)\right)\right). \tag{2}$$

Finally, the sum of the outputs $\mathbf{f}_a =$ from all experts weighted by their corresponding weights in $\mathbf{w}_a$ yields the predicted logits $f = \mathrm{Softmax}(\mathbf{w}_a \mathbf{f}_a)$ after normalization by Softmax function.

To ensure that the model can train each expert in a balanced manner, we use the load balancing loss $\mathcal{L}_g$ to constrain the model to train each expert equally, which is calculated as the squared coefficient of variation of the load $g_L$ and importance $g_I$ of experts. With $\lambda$ denotes the scaling factor, our loss $\mathcal{L}$ comprises the classification loss $\mathcal{L}_{cls}$ and the $\mathcal{L}_g$, which is formalized as:

$$\mathcal{L} = \mathcal{L}_{cls} + \lambda\mathcal{L}_g = -\sum_{i=1}^K y_i \log(f_i) + \lambda\left(\frac{\sigma^2(g_I)}{\mu^2(g_I)} + \frac{\sigma^2(g_L)}{\mu^2(g_L)}\right). \tag{3}$$

### 4.3 REJECTING UNKNOWN SAMPLES

In OSR, the scoring function $\mathcal{S}$ is used to determine whether a sample is known. If the score of the sample is higher than the threshold $\tau$, it is classified as a known sample, otherwise, it is rejected as an unknown sample. We adopt the highest probability predicted by the model across all known classes as the score. The threshold $\tau$ is set to ensure 95% of the known samples are correctly classified.

$$\hat{y}_i = \begin{cases} \mathrm{argmax}_{i=1}^{C_K} f_i, & \text{if } \mathcal{S}(\mathbf{x}_1, \mathbf{x}_2) = \max_{i=1}^{C_K} f_i \geq \tau \\ U, & \text{else} \end{cases}. \tag{4}$$

In equation 4, the model outputs the classification results of known samples that have scores higher than $\tau$, while rejects samples that have scores below the threshold $\tau$ as unknown samples.

## 5 EXPERIMENT

### 5.1 SETUP

**Datasets.** We used various multimodal data to evaluate the proposed method under the MMOSR settings, including IMAGE-TEXT: Food-101 (Wang et al., 2015), a large collection of food images paired with recipe texts across 101 categories, and Flower-102 (Nilsback & Zisserman, 2008), which contains 102 flower categories with detailed image and descriptive text, AUDIO-VISUAL: CREMA-D (Cao et al., 2014), an emotion recognition dataset where actors express different emotions through spoken sentences, and RGB-DEPTH IMAGES: SUN RGB-D (Song et al., 2015), a large-scale dataset contains RGB and depth images from diverse indoor environments for scene understanding.

**Implementation details.** We employed ResNet34 (He et al., 2016) as the RGB / depth images and audio encoder while employing attention-based Bi-LSTM (Zhou et al., 2016) as the text encoder.

Table 2: Comparison of open set recognition results on different methods. The best results are marked in **bold** and gains ($\Delta$ ↑/↓) were calculated within each setting group.

| Method | Food-101 | | Flower-102 | | CREMA-D | | SUN RGB-D | |
|---|---|---|---|---|---|---|---|---|
| | AUROC | OSCR | AUROC | OSCR | AUROC | OSCR | AUROC | OSCR |
| *Single-modal OSR methods* | | | | | | | | |
| ARPL (TPAMI'22) | 62.19 | 41.48 | 66.37 | 54.80 | 53.09 | 17.15 | 62.63 | 40.31 |
| OpenAUC (NeurIPS'22) | 62.21 | 43.72 | 65.74 | 48.85 | 53.90 | 19.95 | 61.46 | 39.73 |
| CSSR (TPAMI'23) | 70.72 | 57.91 | 68.01 | 55.96 | 55.02 | 22.53 | 62.87 | 41.88 |
| ASH (ICLR'23) | 68.41 | 55.76 | 70.40 | 63.15 | 60.12 | 33.64 | 65.35 | 47.52 |
| *Multimodal fusion methods* | | | | | | | | |
| TMC (TPAMI'23) | 88.78 | 86.07 | 69.33 | 60.15 | 66.41 | 52.79 | 64.14 | 46.05 |
| GQA (EMNLP'23) | 89.84 | 85.92 | 68.81 | 59.61 | 67.19 | 53.07 | 64.53 | 46.68 |
| MLA (CVPR'24) | 91.44 | 87.78 | 73.76 | 64.47 | **67.83** | **57.50** | 62.95 | 45.02 |
| **MRN** | **92.16** | **89.16** | **76.23** | **69.70** | 66.78 | 57.32 | **65.72** | **47.53** |
| *Gain* ($\Delta$) | (0.72↑) | (1.38↑) | (2.47↑) | (5.23↑) | (1.05↓) | (0.18↓) | (0.37↑) | (0.01↑) |
| *Multimodal fusion with OSR methods* | | | | | | | | |
| ARPL-ADD | 90.45 | 86.21 | 68.10 | 56.39 | 64.12 | 51.90 | 63.70 | 45.21 |
| ARPL-CAT | 90.92 | 86.35 | 67.63 | 57.37 | 63.34 | 49.35 | 64.19 | 44.67 |
| ARPL-GQA | 90.76 | 84.31 | 69.93 | 59.57 | 63.30 | 53.07 | 63.34 | 45.12 |
| **ARPL-MRN** | **91.24** | **86.76** | **72.68** | **65.54** | **64.37** | **56.79** | **64.50** | **46.08** |
| *Gain* ($\Delta$) | (0.32↑) | (0.41↑) | (2.75↑) | (5.97↑) | (0.25↑) | (3.72↑) | (0.31↑) | (0.87↑) |
| CSSR-ADD | 91.41 | 87.94 | 71.52 | 63.33 | 64.54 | 53.81 | 64.95 | 46.32 |
| CSSR-CAT | 91.56 | 87.53 | 72.48 | 64.91 | 64.58 | 53.77 | 65.18 | 46.36 |
| CSSR-GQA | 91.23 | 87.37 | 71.82 | 63.69 | 65.22 | 54.26 | 65.01 | 45.87 |
| **CSSR-MRN** | **91.82** | **88.51** | **74.02** | **66.21** | **66.86** | **56.90** | **65.37** | **46.59** |
| *Gain* ($\Delta$) | (0.26↑) | (0.57↑) | (1.54↑) | (1.30↑) | (1.64↑) | (2.64↑) | (0.19↑) | (0.23↑) |

We set the expert number $N$ to 15 and the selected number $K$ to 4. To avoid introducing unknown information, we trained all models from scratch only with known classes.

**Baselines.** We set the baselines based on the existing OSR method and multimodal fusion method. For the single modality, we used advanced OSR and out-of-distribution detection methods (Chen et al., 2022; Wang et al., 2022; Huang et al., 2023; Djurisic et al., 2023). For the multimodal fusion methods, we used TMC (Han et al., 2023), GQA (Ainslie et al., 2023), and MLA (Zhang et al., 2024).

### 5.2 RESULTS ON OPEN SET RECOGNITION

Following the standard OSR setting (Chen et al., 2022; Huang et al., 2023), we randomly selected ten classes as known for the Food-101, Flower-102, and SUN RGB-D datasets. For the six-class dataset CREMA-D, we used three emotion classes as known. In addition to the AUROC metric, we adopted OSCR, which evaluates both closed-set and open-set classification performance. All reported results for other methods were reproduced by us using the official code and the same encoder.

**(1) MRN consistently demonstrates exceptional MMOSR performance across various datatypes.** Results in Table 2 indicate that even in more challenging scenarios, such as with increased unknown classes and higher closed-set similarity, MRN maintains strong performance in both closed-set classification and unknown detection. These results underscore the robustness and effectiveness of MRN in tackling the MMOSR task, further validating its utility in real-world multimodal settings.

**(2) MRN extracts multimodal representations comprehensively for the MMOSR tasks.** We integrated OSR methods ARPL (Chen et al., 2022) and CSSR (Huang et al., 2023) with various multimodal fusion approaches and the proposed MRN, then compared the performance in Table 2. MRN has more significant improvements on OSR methods than other fusion strategies with its ability to capture crucial multimodal features that other fusion methods miss mitigates fusion degradation.

**(3) MRN establishes effectiveness across varying scenarios, even outperforming pre-trained multimodal models.** We conducted experiments with numbers of known classes from 5 to 80 to represent different scenarios on the Food-101 dataset. Since it is an image-text dataset, we included

Table 3: Comparison of open set recognition results on different methods under varying scenarios. The best results are marked in **bold** and gains ($\Delta \uparrow/\downarrow$) were calculated within each group.

| Method | # of known classes / # of unknown classes | | | | | | | |
| | 5 / 96 | | 20 / 81 | | 50 / 51 | | 80 / 21 | |
| | AUROC | OSCR | AUROC | OSCR | AUROC | OSCR | AUROC | OSCR |
|---|---|---|---|---|---|---|---|---|
| *Single-modal OSR methods* | | | | | | | | |
| ARPL (TPAMI'22) | 71.88 | 61.91 | 69.94 | 53.28 | 71.00 | 51.96 | 70.83 | 53.52 |
| CSSR (TPAMI'23) | 75.83 | 67.10 | 72.49 | 60.68 | 75.10 | 63.59 | 73.90 | 60.29 |
| ASH (ICLR'23) | 75.24 | 66.64 | 71.34 | 54.41 | 71.05 | 50.94 | 69.22 | 47.44 |
| *Zero-shot / 16-shot fine-tune large-scale pretrained model* | | | | | | | | |
| CLIP (ICML'21) | 90.20 | 86.23 | 84.15 | 73.04 | 79.58 | 65.36 | 73.95 | 57.67 |
| CoOp (IJCV'22) | 86.75 | 83.88 | 79.05 | 68.01 | 73.75 | 58.93 | 68.61 | 52.03 |
| MaPLe (CVPR'23) | 87.16 | 84.47 | 78.63 | 68.11 | 75.47 | 61.24 | 70.23 | 54.47 |
| *Multimodal fusion methods* | | | | | | | | |
| TMC (TPAMI'23) | 95.17 | 92.30 | 85.19 | 83.63 | 78.87 | 84.42 | 83.16 | 82.02 |
| GQA (EMNLP'23) | 95.31 | 92.51 | 91.66 | 86.79 | 90.01 | 85.15 | 89.46 | 82.51 |
| MLA (CVPR'24) | 97.07 | 93.82 | 91.81 | 86.56 | 91.37 | 85.80 | 89.84 | 82.07 |
| **MRN** | **97.30** | **94.86** | **94.80** | **88.24** | **92.89** | **86.23** | **91.83** | **83.30** |
| *Gain* ($\Delta$) | (0.23↑) | (1.04↑) | (2.99↑) | (1.45↑) | (1.52↑) | (0.43↑) | (1.99↑) | (0.79↑) |
| *Multimodal fusion with OSR methods* | | | | | | | | |
| CSSR-ADD | 96.32 | 94.11 | 92.24 | 88.41 | 91.95 | 86.68 | 88.93 | 83.61 |
| CSSR-CAT | 96.55 | 94.29 | 92.40 | 88.64 | 90.51 | 86.03 | 88.71 | 83.25 |
| CSSR-GQA | 96.13 | 93.86 | 92.35 | 88.51 | 91.14 | 86.20 | 88.60 | 83.08 |
| **CSSR-MRN** | **96.81** | **94.69** | **92.60** | **88.93** | **92.03** | **86.88** | **90.14** | **84.19** |
| *Gain* ($\Delta$) | (0.26↑) | (0.40↑) | (0.20↑) | (0.29↑) | (0.08↑) | (0.20↑) | (1.21↑) | (0.58↑) |

the pre-trained image-text model CLIP (Radford et al., 2021) and fine-tuning methods (Khattak et al., 2023; Zhou et al., 2021) as baselines.

Table 3 shows that MRN outperforms across scenarios, excelling in MMOSR tasks. In contrast, large-scale pretrained models designed for general image-text classification, struggle with specific downstream tasks and noisy datasets (Zhou et al., 2021). Their reliance on limited prompts hampers full use of the text modality, highlighting the need for tailored methods like MRN.

## 5.3 FURTHER ANALYSIS

### 5.3.1 ABLATION STUDY

To analyze the effectiveness of each fusion module, we conducted ablation experiments under the open set recognition setting and results are recorded in Table 4.

**The mutually enhanced module effectively reactivates representations and enables cross-modal interaction to improve model performance.** It is worth noting that the improvement is even more pronounced in relatively difficult tasks. Comparing the results of only using $\mathcal{C}_1$ or only using $\mathcal{C}_2$, we considered that when images serve as queries, leverage their richer visual information, leading to improved alignment with textual descriptions, thus enhancing overall performance.

Table 4: Ablation study on the fusion modules. The ✓indicates using the module and the first line refers to the results obtained only with encoders and adaptive fusion.

| Module | | Food-101 | | Flower-102 | |
| $\mathcal{C}_1$ | $\mathcal{C}_2$ | AUROC | ACC | AUROC | ACC |
|---|---|---|---|---|---|
| | | 89.93 | 90.81 | 74.28 | 82.51 |
| ✓ | | 90.52 | 91.45 | 74.83 | 83.05 |
| | ✓ | 91.31 | 92.03 | 75.61 | 83.77 |
| ✓ | ✓ | **92.16** | **92.35** | **76.23** | **84.10** |

### 5.3.2 SENSITIVITY OF HYPERPARAMETERS

To evaluate the sensitivity of the model on the total number $E$ and selected number $K$ of experts, we conducted experiments on Food-101 and Flower-102 datasets under the open set recognition setting.

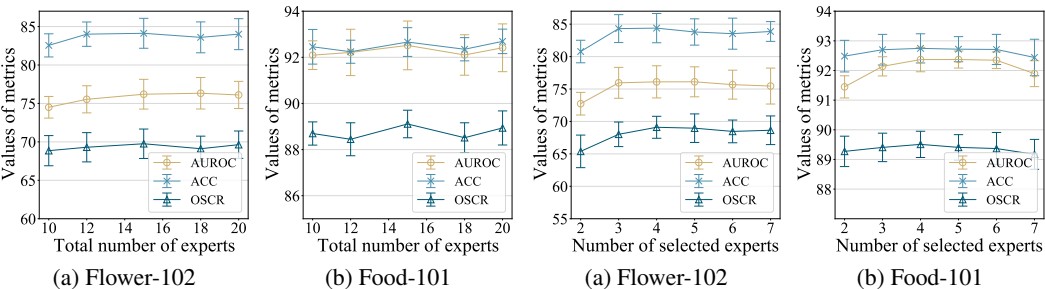

|                  |                  |                  |                  |
| (a) Flower-102   | (b) Food-101     | (a) Flower-102   | (b) Food-101     |

Figure 4: The changes in multiple metrics as the total number $E$ of experts is varying.

Figure 5: The changes in multiple metrics as the selected number $K$ of experts is varying.

**MRN maintains outstanding performance with different total numbers of experts and selected numbers.** Figure 4 shows that with the total number of experts varying from 10 to 20, the overall performance remains stable, indicating that the model is insensitive to hyperparameter expert number $E$. Considering the top-$K$ strategy of the gating network, we varied $K$ from 2 to 7 to test its impact on the model. Figure 5 shows that the MRN maintains consistency except for slight performance drop when adopting either extremely low or high $K$.

### 5.3.3 VISUALIZATION AND DISCUSSION

**MRN exhibits stronger discriminative ability and more comprehensive representations.** We employed t-SNE to visualize the feature scatter of our method in Figure 6. Unknown representations (red dots) exhibit considerable distance from each known class, with known clusters dispersed across the entire space, validating that MRN can distinguish unknown samples from similar known ones. We present the Grad-CAM results of the baseline and MRN in Figure 7. The representations of MRN are more accurate and comprehensive than the baseline, validating its superior representation capability.

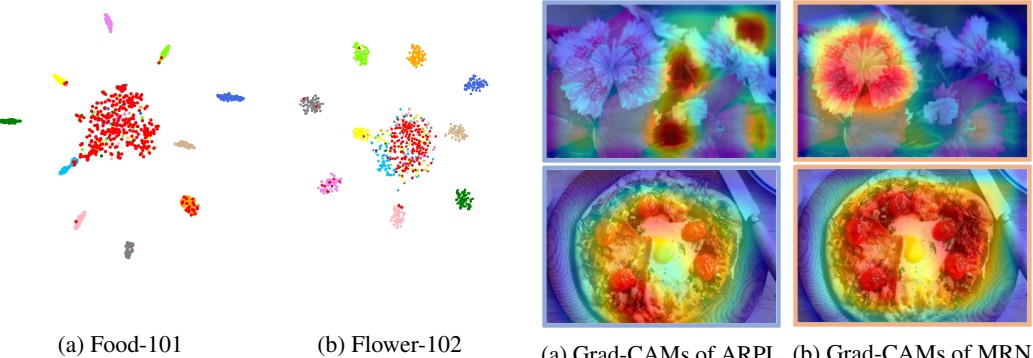

|                 |                 |                         |                       |
| (a) Food-101    | (b) Flower-102  | (a) Grad-CAMs of ARPL   | (b) Grad-CAMs of MRN  |

Figure 6: The scatters of downscaled features with t-SNE on different datasets.

Figure 7: Grad-CAMs of ARPL and MRN on Food-101 and Flower-102 datasets.

**The necessity of designing specific methods for the MMOSR task.** Comparing the performance of multimodal fusion methods with that of methods combined with OSR techniques, we can observe that the constraints imposed by existing OSR methods do not result in consistent improvements on multimodal features. This indicates that the unique challenges posed by MMOSR need to be addressed through designs tailored to its characteristics.

**The significance of MRN in the MLLMs era.** No matter how much data a model has learned, it must be able to clearly identify unknown samples outside the current task. Although large-scale pretrained models have significantly improved performance, they still face substantial challenges in recognizing what they truly don't know. We hope the academic community will further explore this task to enhance the human-like perception capabilities of large models (Cheng et al., 2024; Rawte et al., 2023).

# 6 CONCLUSION

We discuss a new MMOSR task in this paper by analyzing its research necessity and main challenges. We first find that introducing OSR to multimodal fusion would lead to fusion degradation, which seriously affects the model's ability to represent multimodal data by the feature suppression of modality and limited representation capability of the model. To address the aforementioned problems, we then propose a multimodal representation reactivation network that yields comprehensive and informative multimodal representations via mutually enhanced fusion and adaptive fusion. The mutually enhanced fusion module uses cross-attention to enable full modality interactions and enhance the representations, while the adaptive fusion module uses the mixture-of-experts to obtain comprehensive representations. We conducted various experiments and verified the effectiveness of the proposed MRN. Results show that, compared to single-modal OSR, multimodal fusion methods, and simple combinations of OSR and multimodal fusion methods, the proposed multimodal fusion method can significantly enhance the learned representations to facilitate OSR.

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

## A  APPENDIX

In the appendix, we provide a more comprehensive exploration of details not covered in the main text. We present the reproducibility of the proposed method, supplemental experiments, the pseudo-code of the proposed method, etc. We discuss the following subjects in detail:

- Reproducibility of the proposed method
- Supplementary experiments
- Overview of the training and test process
- Experimental details
- Visualizations

## B  REPRODUCIBILITY OF THE PROPOSED METHOD

In this section, we explain the reproducibility of our paper, including the accessibility code and datasets. We also present a comprehensive description of the dataset contents and elaborate on the data processing methods employed.

## B.1 THE CODE OF OUR METHOD

We have submitted our code files in the supplementary materials, which include the necessary files for data processing, model construction, and training. The experimental results presented in the paper can be reproduced by running this code. The code will be made public once the paper is accepted, which allows the researcher to reproduce the results of this paper and conduct further development.

## B.2 THE ACCESSIBILITY AND DETAILS OF THE DATASETS

Since our dataset is built upon existing datasets, it will be made publicly available in sync with the publication of our paper for use by other researchers.

### B.2.1 DETAILS OF THE DATASETS

*(1) Text-image multimodal datasets*

- Flower-102 (Nilsback & Zisserman, 2008): The flower dataset consists of 8,189 images of flowers from 102 different species. Each species has between 40 to 258 images. It is widely used for tasks such as image classification and fine-grained visual recognition.
- UPMC FOOD-101 (Wang et al., 2015): The Food dataset is a large-scale dataset for multi-modal learning tasks, especially designed for food recognition. It contains approximately 100,000 image-text pairs across 101 food categories. Each category represents images of a specific type of food along with detailed textual descriptions.
- Caltech-UCSD Birds (CUB-200) (Wah et al., 2011): The CUB-200 dataset is a popular dataset for fine-grained image classification. It contains 11,788 images of 200 bird species, each annotated with species labels, bounding boxes, and part locations.

*(2) Audio-visual multimodal dataset*

- CREMA-D (Cao et al., 2014) is an emotional multimodal dataset consisting of 7,442 original clips from 91 actors (48 male, 43 female) aged 20 to 74, representing various races and ethnicities. The actors expressed six emotions (Anger, Disgust, Fear, Happy, Neutral, and Sad) through a selection of 12 sentences.

*(3) RGB-depth image multimodal dataset*

- SUN RGB-D (Song et al., 2015) is a large-scale multimodal dataset designed for scene understanding and object detection in indoor environments. It contains over 10,000 RGB-D images, which include both RGB (color) data and depth information. For scene classification, we used the 19 classes with more than 80 images.

**The construction of MMOSR datasets** The Flower-102 dataset shares similarities with the CUB-200 dataset as the images primarily consist of the main objects and exhibit minimal background. The associated text provides descriptions of the representations of flowers or birds, without explicitly mentioning the specific object class names. A representative sample from the flower dataset and the CUB-200 dataset, including both the image and the corresponding text, is depicted in Figure 8a and Figure 8b, respectively. The text descriptions in the Food dataset are compiled from recipe descriptions, removing category names is not a specific focus. For example, in Figure 8c, the corresponding label "creme brulee" and "frozen yogurt" are in included in the text description.

For the Flower-102 and CUB-200 datasets, dictionaries are provided by Reed et al. (2016) that encompass all the words found in their respective text descriptions. Each word in the dictionary is assigned a unique number for text encoding purposes. In the case of the Food dataset, a pre-existing dictionary is not provided. Hence, we constructed our own dictionary by incorporating all the words present in the text descriptions. To ensure the effectiveness of the dictionary, we removed words that occur fewer than five times. We employed the bag-of-words modeling approach to convert the textual data into numerical codes. As depicted in Figure 8, there is a variation in the text length across different datasets. In the case of the Flower-102 and CUB-200 datasets, the text length is limited to 150 words. For the Food dataset, the text length is restricted to 10 words.

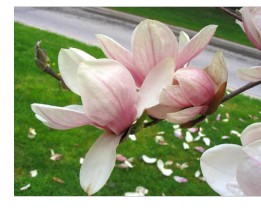

these pretty flowers has pink and white petals with pointed edges. this flower has overlapping layers of white petals with pale pink pointed tips. this flower is pink and white in color, with petals that are oval shaped. this flower has petals that are white with pink shading. this flower has white petals that have pink veining on the back side. this flower has pink and white petals attached to a brown pedicel. this flower is white and pink in color, and has petals that are oval shaped. this flower has long white petals streaked with pink, with tapered tips. this flower has petals that are white and has pink shading. this pale pink flower with hints of white sits in a group of other flowers at the end of a brown branch.

(a) A sample from the Flower-102 dataset.

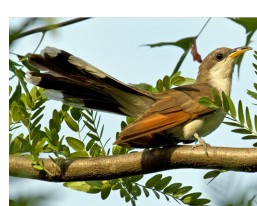

a bird that has a long tail, and multiple brown tones and a white underbelly. this small gray bird has a large tail that is sticking somewhat upwards. this bird has a brown crown, a white throat, and a yellow bill. this bird has a white breast and a really long tail. this brown bird has ornate plumage, neatly separated between dark orange, brown, and white. this bird is brown with white and has a long, pointy beak. this bird has a white belly and breast, a grey back and an orange wing and beak. this primarily brown and white bird is generally small, but has long inner and outer retrices and relatively large eyes. this bird has very long retrices and orange on its wings. this bird has a pointed yellow bill, with a white breast.

(b) A sample from the CUB-200 dataset.

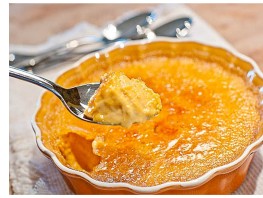

Pumpkin Creme Brulee Recipe - Cooking | Add a Pinch | Robyn Stone,creme_brulee

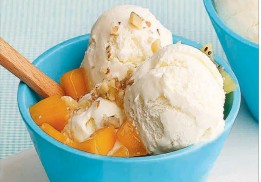

Easy-Peasy Vanilla Frozen Yogurt | Recipes | Yummy.ph - the online source for easy Filipino recipes,frozen_yogurt

(c) Samples from the Food-101 dataset.

Figure 8: Samples from text-image dataset.

After processing the data for both image and text modalities, we divided the known and unknown classes according to the open set recognition setup and used the known classes to train the model.

### B.3 MORE DETAILS OF THE PROPOSED METHOD

To ensure that each expert receives adequate training and to adhere to the load balancing proposed by Shazeer et al. (2016), we set the loss $\mathcal{L}_g$. The expert importance $g_I$ mentioned in the main text is obtained by calculating the sum of the weight values $\mathbf{w}_a$ allocated to the expert within a batch, and the expert load $g_L$ represents the degree of the tendency of the gating network to assign tasks to this expert. With $\mathbf{z}_c$ denotes the feature outputted by cross attention and the Gaussian noise $\epsilon \sim \mathcal{N}(0, \mathbf{I})$, we determine the relative ranking of the scores for different experts as

$$\mathbf{s} = \mathbf{W}_g \mathbf{z}_c + \epsilon \cdot \mathrm{Softplus}\left(\mathbf{W}_{\mathrm{noise}} \mathbf{z}_c\right) \tag{5}$$

For the $e$-th expert, the probability $P(\mathbf{z}_c, e)$ that it is selected in top-$K$ experts is calculated as equation 6, where $\mathbf{s}_e^K$ represents the $K$-th largest value in the $\mathbf{s}$, excluding the $e$-th expert. $\Phi$ is the cumulative distribution function (CDF) of the standard normal distribution and $B$ denotes the batch size. The load $g_L^e = \sum_{b=1}^{B} P(\mathbf{z}_c^b, e)$ and the importance $g_I^e = \sum_{b=1}^{B} \mathbf{w}_a^{e,b}$.

$$P(\mathbf{z}_c, b) = \Phi\left(\frac{(\mathbf{W}_g)_b \mathbf{z}_c - \mathbf{s}_b^K}{\mathrm{Softplus}((\mathbf{W}_{\mathrm{noise}} \mathbf{z}_c)_b)}\right) \tag{6}$$

## C SUPPLEMENTARY EXPERIMENTS

### C.1 DEEPER ANALYSIS OF THE FUSION DEGRADATION AND REPRESENTATION INSUFFICIENCY

To further illustrate the issue of fusion degradation in existing methods, we conducted experiments on two datasets with five or ten known classes and plotted the difference between the optimal classification performance of single modalities and the fused results in Figure 9. The calculation of $\triangle \mathrm{ACC}_c$ is as follows:

$$\triangle \mathrm{ACC}_c = \max\left(\mathrm{ACC}_c^{\mathrm{Image\text{-}OSR}}, \mathrm{ACC}_c^{\mathrm{Text\text{-}OSR}}\right) - \mathrm{ACC}_c^{\mathrm{Fusion\text{-}OSR}}, c \in \{1, ..., C_K\}. \tag{7}$$

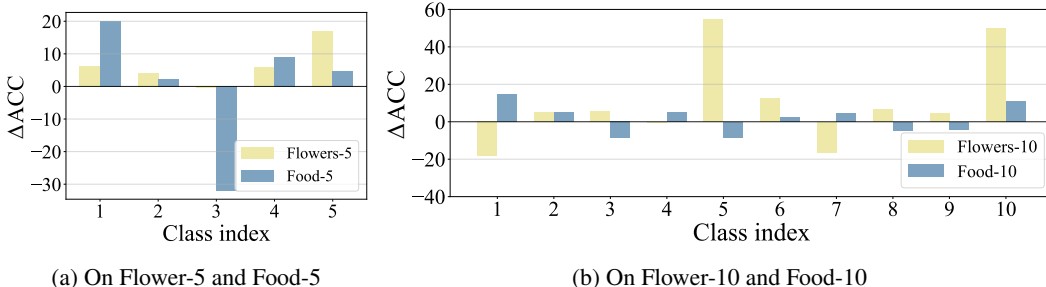

(a) On Flower-5 and Food-5        (b) On Flower-10 and Food-10

Figure 9: The ACC difference of each known class between Image-OSR, Text-OSR, and Fusion-OSR. The '-5' and '-10' denote using 5 or 10 classes from the corresponding dataset as known classes.

Table 5: Comparison of out-of-distribution detection results on different methods. The best results are marked in **bold**.

| Method | InD: Food-101 / OOD: Flower-102 | | | | InD: Food-101 / OOD: CUB-200 | | | |
|---|---|---|---|---|---|---|---|---|
| | TNR | DTACC | AUROC | OSCR | TNR | DTACC | AUROC | OSCR |
| *Single-modal OSR methods* | | | | | | | | |
| ARPL (TPAMI'22) | 15.58 | 70.45 | 75.85 | 52.11 | 19.16 | 73.81 | 78.30 | 51.44 |
| CSSR (TPAMI'23) | 19.39 | 71.07 | 77.42 | 53.06 | 21.77 | 75.34 | 81.62 | 51.83 |
| ASH (ICLR'23) | 24.12 | 72.79 | 80.49 | 49.53 | 27.28 | 76.76 | 83.92 | 50.82 |
| *Multimodal fusion methods* | | | | | | | | |
| TMC (TPAMI'23) | 61.22 | 90.70 | 94.02 | 83.97 | 68.12 | 93.18 | 95.23 | 85.74 |
| GQA (EMNLP'23) | 61.38 | 91.06 | 94.35 | 84.48 | 67.74 | 93.27 | 95.31 | 86.02 |
| MLA (CVPR'24) | 63.97 | 91.45 | 94.68 | 85.39 | 69.44 | 93.50 | 95.96 | **86.75** |
| MRN | **65.38** | **92.14** | **95.42** | **86.34** | **70.01** | **93.99** | **96.27** | 86.32 |
| *Gain* (Δ) | (1.41↑) | (0.69↑) | (0.74↑) | (0.95↑) | (0.57↑) | (0.49↑) | (0.31↑) | (0.43↓) |

'Image-OSR' and 'Text-OSR' refer to the results of applying OpenAUC (Wang et al., 2022) on only the image modality or the text modality, respectively. 'Fusion' means using addition as a multimodal fusion method without applying OSR methods. 'Fusion-OSR' means combining the multimodal fusion and OSR methods.

Results show that in MMOSR, the fusion of different modalities is beneficial for a small number of known classes but not practical for most known classes. We attribute this phenomenon to the degradation of fused representations relative to single-modality representations, leading to decreased classification performance when using the fused representations.

## C.2   RESULTS ON OUT-OF-DISTRIBUTION DETECTION

Out-of-distribution (OOD) detection identifies samples from distributions different from the training set. We used the entire Food-101 dataset as the in-distribution (InD) dataset and the Flower-102 or CUB-200 serving as the OOD dataset. CUB-200 (Wah et al., 2011) is a fine-grained image-text multimodal dataset focused on 200 bird species. Following (Chen et al., 2022; Huang et al., 2023), we additionally used TNR and DTACC as evaluation metrics. TNR measures the proportion of true negatives among all negatives, evaluated when the True Positive Rate (TPR) is fixed at 95% with threshold $\tau$. DTACC measures the highest classification accuracy across thresholds.

**MRN effectively discriminates OOD samples from various distributions.** Results in Table 5 indicate that MRN has notable improvements over both single-modality OSR and multimodal methods. Although there is a slight decrease in OSCR for the CUB-200 dataset, MRN overall achieves strong detection accuracy and excels in recognizing out-of-distribution samples across various datasets.

Table 6: The performance of the MRN against different openness on the Food-101 and Flower-102 datasets. The number of known classes is fixed at 10 while the number of unknown classes is varying.

| Datasets | $C_U$ | Openness | TNR | AUROC | DTACC | AUIN | AUOUT | OSCR |
|---|---|---|---|---|---|---|---|---|
| | 10 | 18.35% | 57.08 | 93.10 | 88.25 | 92.37 | 92.36 | 89.72 |
| | 15 | 24.41% | 56.91 | 92.95 | 87.93 | 91.08 | 94.83 | 90.37 |
| | 20 | 29.29% | 57.06 | 93.21 | 87.68 | 90.43 | 95.22 | 90.67 |
| Food-101 | 30 | 36.75% | 57.66 | 93.20 | 88.52 | 87.24 | 96.81 | 90.55 |
| | 40 | 42.26% | 57.16 | 93.10 | 88.12 | 83.48 | 97.45 | 90.37 |
| | 60 | 50.00% | 55.96 | 92.88 | 88.14 | 76.36 | 98.22 | 89.89 |
| | 80 | 55.28% | 54.67 | 92.50 | 87.62 | 76.18 | 98.10 | 89.85 |
| | 10 | 18.35% | 32.84 | 86.55 | 81.00 | 94.60 | 66.10 | 83.90 |
| | 15 | 24.41% | 31.95 | 84.93 | 79.83 | 90.76 | 68.44 | 81.54 |
| | 20 | 29.29% | 31.07 | 85.02 | 79.61 | 85.94 | 75.44 | 80.97 |
| Flower-102 | 30 | 36.75% | 32.25 | 84.43 | 78.65 | 82.71 | 80.24 | 81.02 |
| | 40 | 42.26% | 30.74 | 82.88 | 77.64 | 79.66 | 86.47 | 79.71 |
| | 60 | 50.00% | 28.64 | 81.84 | 75.78 | 73.43 | 93.22 | 77.33 |
| | 80 | 55.28% | 25.97 | 82.45 | 78.61 | 69.06 | 95.62 | 76.12 |

Table 7: Analysis of the effectiveness of the adaptive fusion component. The first row indicates using only a linear layer as the classification head, without the adaptive fusion module. $E$=15, $K$=4 is our original setting in main content.

| | | # of known classes / # of unknown classes | | | | | | | | | | |
|---|---|---|---|---|---|---|---|---|---|---|---|---|
| $E$ | $K$ | 10 / 91 | | | 20 / 81 | | | 50 / 51 | | | 80 / 21 | | |
| | | AUROC | ACC | OSCR | AUROC | ACC | OSCR | AUROC | ACC | OSCR | AUROC | ACC | OSCR |
| / | / | 91.23 | 92.22 | 88.17 | 93.77 | 88.94 | 86.88 | 91.34 | 86.91 | 85.02 | 90.54 | 84.83 | 80.84 |
| 1 | 1 | 91.44 | 92.38 | 88.51 | 94.02 | 89.37 | 87.10 | 91.73 | 87.68 | 85.35 | 91.06 | 85.31 | 81.15 |
| 15 | 1 | 90.83 | 92.16 | 87.91 | 93.44 | 88.93 | 86.62 | 90.87 | 86.26 | 83.58 | 89.28 | 84.86 | 80.59 |
| 15 | 4 | 92.16 | 92.80 | 89.16 | 94.80 | 90.18 | 88.05 | 92.89 | 88.12 | 86.23 | 91.83 | 85.73 | 83.30 |

## C.3 EXPERIMENTS ON PERFORMANCE AGAINST OPENNESS

Openness (Scheirer et al., 2012) is defined by equation 8, where $C_K, C_U$ denote the number of known and unknown classes during test. Higher openness indicates a greater likelihood of encountering unknown classes, while more known classes increase classification difficulty.

$$Openness = 1 - \sqrt{(2 * C_K)/(2 * C_K + C_U)}. \tag{8}$$

We conducted out experiments on the Food-102 and Flower-102 dataset with a fixed number of 10 known classes while varying the number of unknown classes $C_U$ from 10 to 80 to investigate different degrees of openness. The openness and metrics are depicted in Table 6. It can be observed that the model demonstrates strong recognition performance across various levels of openness, indicating the robustness of the proposed method.

## C.4 ABLATION STUDY ON THE ADAPTIVE FUSION MODULE.

**The adaptive fusion module enhances the performance of the model, helping it better handle complex classification tasks.** We conducted an ablation study of the adaptive fusion module and added a setting using only one expert. We experimented with this module as a basic linear layer ($E$=0, $K$=0), using one multilayer perceptron expert ($E$=1, $K$=1), and setting 15 experts but selecting only one expert ($E$=15, $K$=1).

Results in Table 7 indicate that adaptive fusion module consistently enhances the model's performance. Using only a linear layer or a single expert reduced the model's robustness, leading to performance degradation in more challenging tasks. The most significant improvements are seen in the AUROC scores, which indicate better overall discrimination capabilities, especially for OSR tasks.

---

**Algorithm 1** The pseudo-code of the MRN training process

---

**Input:** Training set $\mathcal{D}_K = \{(\mathbf{x}_i^1, \mathbf{x}_i^2, y_i)\}_{i=1}^n$, total number of experts $E$, selected number $K$
**Output:** Trained model parameters $\theta$
1: Initialize model parameters $\theta$ randomly
2: **repeat**
3:      Input a batch of samples $\mathbf{x}_1, \mathbf{x}_2$ into encoders and obtain representations $\mathbf{z}_1 = \mathcal{E}_1(\mathbf{x}_1)$, $\mathbf{z}_2 = \mathcal{E}_2(\mathbf{x}_2)$
4:      Obtain mutually enhanced representations $\mathbf{z}_1^c = \mathcal{C}_1(\mathbf{z}_1, \mathbf{z}_2, \mathbf{z}_2), \mathbf{z}_2^c = \mathcal{C}_2(\mathbf{z}_2, \mathbf{z}_1, \mathbf{z}_1)$
5:      Concatenate enhanced text representations and image representations $\mathbf{z}_c = \mathbf{z}_1^c \oplus \mathbf{z}_2^c$
6:      Input $\mathbf{z}_c$ into expert network $\mathcal{N}$ and obtain the predictions $\mathbf{f}_a = (\mathcal{N}(\mathbf{z}_c))$
7:      Input $\mathbf{z}_c$ into gating network $\mathcal{G}$ and obtain the adaptive weights of experts $\mathbf{w}_a$ as equation 2
8:      Obtain the adaptively fused prediction $f = \text{Softmax}(\mathbf{w}_a \mathbf{f}_a)$
9:      Compute classification loss $\mathcal{L}_{cls}$ and load balancing loss $\mathcal{L}_g$
10:      Calculate the loss $\mathcal{L} = \mathcal{L}_{cls} + \lambda \mathcal{L}_g$
11:      Backpropagation the loss $\delta \leftarrow \nabla \mathcal{L}$ and update the parameters $\theta \leftarrow \theta + \eta \delta$
12: **until** *Convergence*

---

## D OVERVIEW OF THE TRAINING AND TEST PROCESS.

**Training phase.** We summarize the training process of MRN in Algorithm 1. The model is trained until convergence. Mutually enhanced representations are extracted in line 4, and adaptively fused representations in line 8. Subsequently, we calculate the classification loss $\mathcal{L}_{cls}$ and load balancing loss $\mathcal{L}_g$ then use the total loss $\mathcal{L}$ to train the model until convergence.

**Test phase.** During test, the initial step comprises computing predictions and scoring outcomes for all samples. The threshold $\tau$ is established by identifying the score value that ensures 95% accuracy in binary classifying known samples. Samples scoring below $\tau$ are rejected, while those scoring equal to or above $\tau$ are considered known, and output the predictions of the model as the final classification results.

## E EXPERIMENTAL DETAILS

### E.1 REVIEWS OF COMPARED BASELINES

In this section, we briefly introduce the baseline methods used for comparison.

*(1) Open set recognition methods*

- ARPL (Chen et al., 2022): The adversarial reciprocal point learning (ARPL) framework aims to reduce the overlap between known and unknown data by modeling the extra space associated with unknown classes and employing reciprocal points. Through adversarial mechanisms, ARPL enhances the model's ability to differentiate unknown classes.
- OpenAUC (Wang et al., 2022): OpenAUC evaluates samples using a concise pairwise approach and checks if the open-set sample ranks higher than the close-set one. This method aligns with Open-Set Recognition goals and mitigates threshold sensitivity for improved open set performance.
- CSSR (Huang et al., 2023): The Class-Specific Semantic Reconstruction (CSSR) leverages class-specific auto-encoders to reconstruct semantic features for each class, reducing open space risk and improving recognition accuracy on both known and unknown classes.
- ASH (Djurisic et al., 2023): The Activation Shaping (ASH) method prunes a large portion of activations (e.g., 90%) during inference and makes minor adjustments to the remaining activations. This approach requires no additional training or network modification, improving OOD detection performance while preserving in-distribution accuracy.

*(2) Multimodal fusion methods*

- TMC (Han et al., 2023): The Trusted Multi-View Classification (TMC) integrates evidence from multiple views using Dirichlet distributions and Dempster-Shafer theory to provide uncertainty-aware and reliable predictions. TMC enhances classification robustness by dynamically combining evidence.

- GQA (Ainslie et al., 2023): As Multi-Query Attention (MQA) notably speeds up decoder inference but may compromise quality, Grouped Query Attention (GQA) was proposed as a streamlined alternative. By using an intermediate number of key-value heads, GQA achieves a similar training speed to MQA while maintaining quality close to multi-head attention.

- MLA (Zhang et al., 2024): MLA addresses the challenge of modality imbalance in multimodal learning by alternating the optimization of unimodal encoders while maintaining cross-modal interaction through a shared head.

*(3) Large-scale pre-trained models*

- CLIP (Radford et al., 2021): CLIP employs contrastive learning to jointly train a vision and language model on large-scale image-text pairs. By aligning visual representations with textual descriptions, CLIP enables zero-shot transfer to diverse downstream tasks, such as image classification and object recognition, without task-specific fine-tuning.

- CoOp (Zhou et al., 2021): CoOp introduces a prompt-learning framework designed to enhance vision-language models like CLIP. By optimizing task-specific prompts rather than using static templates, CoOp improves performance in few-shot learning scenarios, enabling more effective adaptation to various downstream tasks.

- MaPLe (Khattak et al., 2023): MaPLe advances prompt learning by simultaneously optimizing text and visual prompts across modalities. This method enhances the alignment of multimodal representations, leading to superior performance in tasks that require integrated understanding of both vision and language, such as visual question answering and image captioning.

## E.2 DETAILS OF METRICS

- Area under the receiver operating characteristic curve (AUROC): The AUROC is a widely used metric for evaluating the performance of binary classification models. It measures the area under the receiver operating characteristic (ROC) curve, which plots the true positive rate (TPR) against the false positive rate (FPR) at various threshold settings.

- Average accuracy (ACC): ACC is the mean of the accuracies for each individual class. It accounts for the performance of the classifier on each class separately and then averages these performances.

- Open set classification rate (OSCR): OSCR measures the rate of correctly rejecting unknown samples while correctly classifying the known classes. With $\tau$ denotes the threshold for rejecting unknown samples and the test set of known samples is denoted as $\mathcal{D}_T^K = \{(\mathbf{x}_i^1, \mathbf{x}_i^2, y_i)\}_{i=1}^{m_k}, y_i \in \{1, ..., C_K\}$, OSCR is calculated as

$$\text{OSCR} = \frac{\sum_{i=1}^M \mathbb{1}\left(\hat{y}_i = y_i \wedge \mathcal{S}(\mathbf{x}_i^1, \mathbf{x}_i^2) \geq \tau\right)}{m_k}.$$

  As the OSCR curve represents the Correct Classification Rate (CCR) at different False Positive Rate (FPR). Following the common setting of Chen et al. (2022), we reported the area under CCR against the FPR curve as OSCR values in our experiments.

- Detection acceptance rate (DTACC): DTACC is a metric designed for evaluating open set recognition models. It measures the average probability that a correctly identified sample is assigned a higher score than any open-set sample.

- Area under the precision-recall curve (AUPR): In scenarios where the dataset is imbalanced (i.e., the number of positive samples is much smaller than the number of negative samples), traditional metrics like accuracy can be misleading. AUPR provides a more nuanced measure that focuses on the performance concerning the positive class. AUIN measures the AUPR where the known samples are considered as the positive class while the AUOUT measures the AUPR with unknown samples being treated as the positive class.

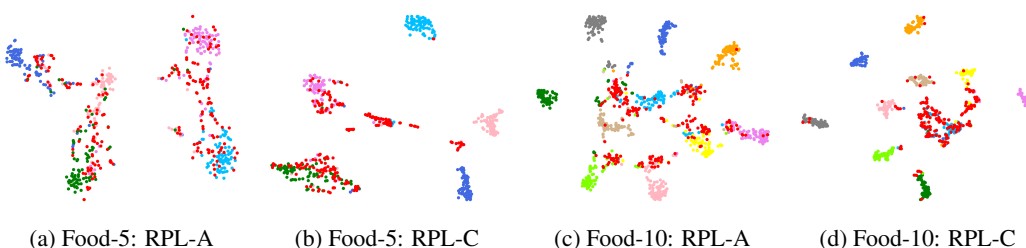

| (a) Food-5: RPL-A | (b) Food-5: RPL-C | (c) Food-10: RPL-A | (d) Food-10: RPL-C |

Figure 10: RPL uses add (-A) or concatenate (-C) to get the t-SNE plots of multiple datasets. The red dots represent unknown samples. The '-5' and '-10' denote using 5 or 10 classes from the corresponding dataset as known classes.

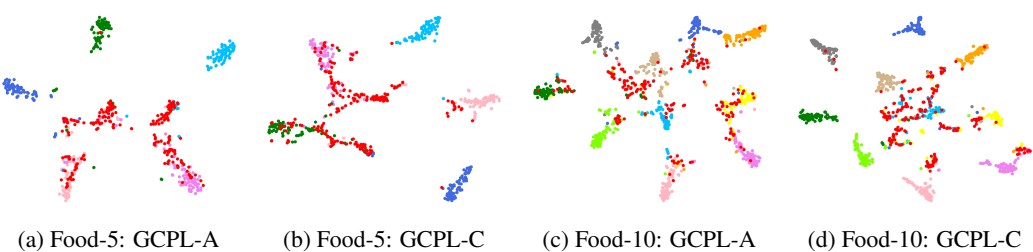

| (a) Food-5: GCPL-A | (b) Food-5: GCPL-C | (c) Food-10: GCPL-A | (d) Food-10: GCPL-C |

Figure 11: GCPL uses add (-A) or concatenate (-C) to get the t-SNE plots of Food-5 and Food-10. The red dots represent unknown samples. The '-5' and '-10' denote using 5 or 10 classes from the corresponding dataset as known classes.

- True Negative Rate (TNR): TNR measures the rate at which the model correctly identifies negatives at a 95% confidence level. A higher TNR indicates better performance of the model in negative prediction, meaning the model makes fewer mistakes in predicting negatives as positives at a 95% confidence level.

# F VISUALIZATIONS

## F.1 VISUALIZATION OF BASELINES

In this section, we present the t-SNE feature scatter plots of additional uni-modal open-set recognition methods combined with multimodal fusion methods to validate the existing issues of over-compression on known classes that we have identified further. We utilized two other OSR methods: RPL (Chen et al., 2020) and GCPL (Yang et al., 2022). RPL enhances open set recognition by using discriminative reciprocal points to distinguish between known and unknown classes, and GRPL is a method that integrates convolutional neural networks with prototype learning to create class-specific prototypes, addressing the OSR tasks by leveraging feature space distances.

From Figure 10 and Figure 11, we can observe that both of them exhibit the issue of over-compression in scenarios with different number of known classes (-5 / -10) when using the multimodal representations, leading to confusion between unknown samples and similar known classes. This validates that the challenge of fusion degradation proposed in this paper is prevalent among OSR methods.

## F.2 VISUALIZATION OF MRN

**MRN can effectively and comprehensively focus on the main subjects in the images for classification.** In Figure 12, we obtain Grad-CAM of multiple models on the Food dataset, and it can be observed that the image attention of MRN is significantly more comprehensive compared to other methods. This ensures a comprehensive focus on the classification subject. This also validates that our method has to some extent addressed the problem of inadequate representational capability in existing open set recognition methods.

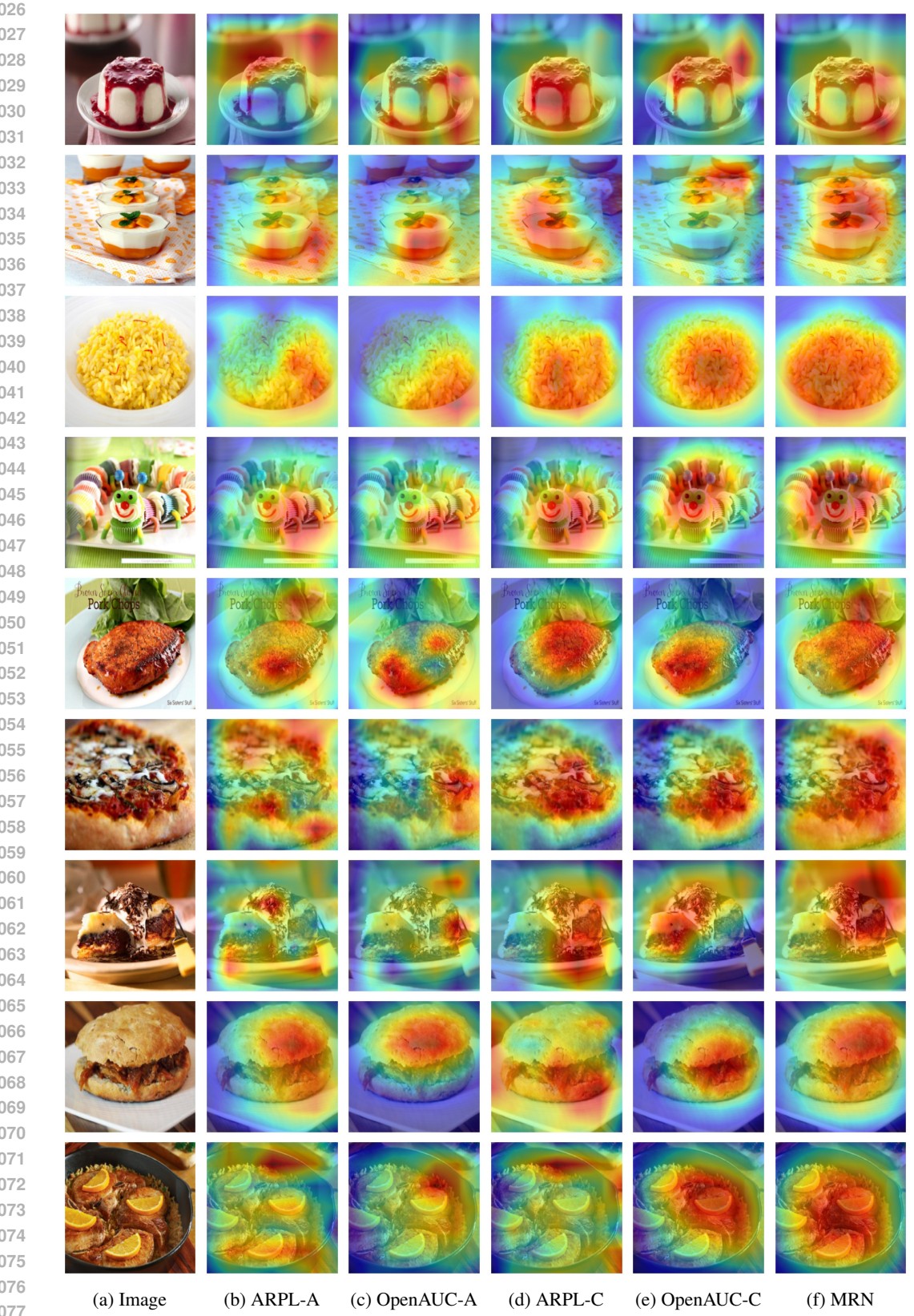

|                |                |                |                |                |                |
| :------------: | :------------: | :------------: | :------------: | :------------: | :------------: |
| (a) Image      | (b) ARPL-A     | (c) OpenAUC-A  | (d) ARPL-C     | (e) OpenAUC-C  | (f) MRN        |

Figure 12: Comparison of Grad-CAMs obtained with different methods on the Food-101 dataset. '-A' and '-C' denote using add or concatenate to fuse the multimodal representations.

