# OpenReview forum: "Towards Multimodal Open Set Recognition"
_ICLR.cc/2025/Conference — Submitted to ICLR 2025_

### Official Review · Reviewer_8qGX · 2024-10-31

**Soundness:** 2
**Presentation:** 3
**Contribution:** 2
**Rating:** 5
**Confidence:** 4

**Summary:**

This submission proposes a multi-modal open-set recognition paradigm (MMOSR) by exending OSR to more practical scenarios with multiple modalities. To exploit multi-modal data, the proposed method focus on multi-modal fusion for OSR, and propose a multimodal representation reactivation network to alleviate fusion degradation. Specifically, two cross-attention modules are deployed and mixure of experts is considered for adaptive fusion. Experiments show the effect.


Since the authors do not provide feedback, I therefore keep my rating.

**Strengths:**

1. Exploring multimodal fusion for OSR is interesting and approaches practical scenarios.
2. The proposal of fusion degradation problem sounds good.

**Weaknesses:**

1. There lacks clear and sufficient analysis about the fusion degradation challenge. In-depth insights about this challenge should be discussed and confirmed.
2. The technical novelty is limited. From the technical idea of the multimodal fusion, I could not learn more about new knowledge. Cross-attention and mixure of experts are very general strategies proved to be effective. However, the proposed method is simple application of such techniques. Therefore, it is unclear why and how these techniques overcome the challenge of fusion degradation.
3. Figure 1 may have no useful information about the proposed method, and even this figure is removed.
4. As figure 3 shows, only two modalities are considered in method design, which may not be general.
5. MoE will cause more computation, which should be taken into account in method selection.

**Questions:**

1. How about the impact of the Gaussian nosie \epslion in Eq. 2 to the final performance? since you claim this is for improving robustness and preventing overfitting. But this is not discussed in experiments.

---

### Official Review · Reviewer_EB2Y · 2024-11-02

**Soundness:** 2
**Presentation:** 3
**Contribution:** 2
**Rating:** 3
**Confidence:** 4

**Summary:**

This paper raises the problem that most research on open set recognition is monomodal, introducing MMOSR (multi-modal open set recognition), and points out that the use of other multi-modal methods may lead to fusion degradation, thus introducing MRN, designing a mutually enhancing fusion module to enable full interaction between different modalities.

**Strengths:**

1.	This paper addresses the open set recognition problem for multimodal data, which can solve more issues in the fields of multimodal and open set recognition.
2.	When performing modality fusion, it identifies the issue of fusion degradation and designs a module to eliminate it.

**Weaknesses:**

1.	The model is overly simplistic. The innovative point of the MRN's adaptive fusion module merely employs an expert mixture model.
2.	When dealing with reject samples (belonging to unknown classes), the method uses softmax values less than the parameter τ, but does not provide a basis for selecting this parameter. If it is a hyperparameter, how was it determined? If it is a learnable parameter, how is it learned?
3.	In the experimental setup section, the experimental environment and other parameters are not specified.
4.	In the experimental results, for the open-set multimodal tasks proposed in this paper, the advantages of CSSR-MRN are not significant for most datasets.

**Questions:**

see weakness

---

> ### Comment · Reviewer_EB2Y · 2024-11-26
>
> The authors do not respond my conerns, so I keep my score.

---

### Official Review · Reviewer_x6GR · 2024-11-03

**Soundness:** 3
**Presentation:** 2
**Contribution:** 3
**Rating:** 5
**Confidence:** 4

**Summary:**

The paper addresses the issue of multi-modal (MM) open-set recognition (OSR).
The authors first discuss simplistic MM fusion techniques and show that they do not improve closed-set ACCuracy or out-of-distribution detection AUROC.
To solve this, they propose to use cross-attention between features of different modalities, and a mixture-of-experts system to further improve performance.
Experiments on various bi-modal datasets show improved performance for the proposed method as compared to related methods.

**Strengths:**

+ The proposed framework is straightforward and should be relatively easy to implement.
+ The experimental evaluation procedure is designed correctly. Particularly, networks seem to be trained on the known classes only, hopefully without pre-training on external datasets that might include the test classes.
+ Results show improved performances over recent baselines.
+ Ablation studies show the contributions of all components.

**Weaknesses:**

While in general the results indicate reasonable improvements, there are some points that need to be resolved before the paper is of sufficient quality to be published:

1. The structure of the document and the description of the proposed method would enjoy improvements:

   a) The organization of the paper can be improved. It is not clear why results are presented in section 3, before discussing details of the method and the evaluation?

   b) For the preliminary experiments in section 3.2, what are the models used to extract features for text and image modalities? What does it exactly mean to "add[ing] representations from both modalities"? If this is this simple element-wise addition, why would this be a reasonable fusion strategy? Why not using score fusion techniques which are typically better suited for MM fusion? What are the two variations Fusion and Fusion-OSR reported in table 1?

   c) Section 4.3 proposes a specific evaluation procedure, but such a procedure is not used in their evaluation. Why do the authors talk about such standard evaluation?

   d) There are many results in table 3 that are neither described nor discussed in the text. Particularly, the final block appears to show variations of the proposed method, which are explained nowhere.

   e) The role of the visualizations in figures 7 and 12 is unclear. What do the authors expect to see? Are these representations for known or unknown samples? To me, all visualizations highlight reasonable locations, I cannot follow the conclusion drawn by the authors that "representations of MRN are more accurate and comprehensive than the baseline". The authors should consider sacrificing these plots (or move all of them to the supplemental) and use the space to discuss their results better.

   f) There is quite a lot of additional information in the appendix, which is not mentioned anywhere in the main paper. The authors should add indications into the main text, and discuss why we see such supplemental material.

   g) In algorithm 1 (in the supplemental material), it is not clear which parameters are included into the set \theta. Specifically, are the parameters of the encoder networks also included in these parameters, or are pre-trained networks used?


2. The evaluation needs to be improved:

   a) The conclusions drawn for the initial experiments are solely valid for the applied OSR and MM methods and might not transfer to other methods. The authors need to acknowledge this and tone down claims such as "open space risk [...] cannot be resolved by simply integrating OSR methods".

   b) Evaluation is based on ACC, AUROC and AUOSCR, all of which are mainly influenced by improved closed-set classification. This is perfectly shown in figures 4 and 5 where AUROC and OSCR values mostly correlate with ACC. A better evaluation would plot OSCR curves [Dhamija2018] which incorporate closed- and open-set evaluation into a single curve. Also, OpenAUC (Wang et al., 2022) discusses why ACC and AUROC are not good metrics to evaluate OSR.

   c) The employed Max-probability baseline OSR technique is very simplistic and not SOTA. For a similarly simply, but much more effective, OSR technique, the authors should rather use the MaxLogits technique [Hendrycks2022].

   d) The ablation study in section 5.3.2 only varies the number of experts in between 10 and 20. It misses showing that the module itself provides improvements over not at all using a mixture of experts. Such results are only presented in the supplemental material.

   e) The visualization of high-dimensional features is done via t-SNE, which has been superseded by UMAP. The authors should consider to use UMAP as well. Representations are not "downscaled", but "dimension-reduced".


3. The mathematical details need to be improved:

   a) In (1), why is the same value z_2 input to function C_1 twice?

   b) Above (2), a value \epsilon is defined, but not used in (2).

   c) Below (2), it is mentioned that the logits f are the output of Softmax, while by definition logits are the inputs to Softmax.

   d) In (3), g_I and g_L are used without providing equations on how these values are computed.

   e) The number of experts is called E in section 4.2.2 and N in section 5.2.


[Dhamija2018] A. R. Dhamija, M. Gunther, and T. E. Boult, "Reducing network agnostophobia," in Advances in Neural Information Processing Systems (NeurIPS), 2018.

[Hendrycks2022] D. Hendrycks et al., "Scaling Out-of-Distribution Detection for Real-World Settings," in International Conference on Machine Learning (ICML), 2022.

**Questions:**

See Weaknesses, particularly points 1 and 2.

---

> ### Comment · Reviewer_x6GR · 2024-11-26
>
> Since the authors provided no feedback, this paper should be rejected.

---

### Official Review · Reviewer_CgBN · 2024-11-03

**Soundness:** 2
**Presentation:** 2
**Contribution:** 1
**Rating:** 5
**Confidence:** 4

**Summary:**

The paper extends traditional open set recognition (OSR) to multimodal data. The paper identifies that current single-modal OSR, fail in multimodal scenarios because of "fusion degradation"—a challenge where the fusion of multiple modalities compresses the representation space too tightly, limiting the model's ability to recognize unknown samples. To address this, the authors propose a Multimodal Representation Reactivation Network (MRN), which reactivates and enhances multimodal representations. The paper demonstrates that MRN achieves better performance than existing methods in terms of accuracy and UROC in various datasets.

**Strengths:**

The paper is easy to follow.
The empirical results show satisfactory improvements on benchmark datasets.
The idea of mutually enhanced fusion is intuitive and well-motivated.

**Weaknesses:**

- Experiment in Table 1: The evaluation in Table 1 appears to be somewhat unfair. Using simple summation to fuse representations from different modalities may not be effective, as the embeddings from two distinct models (modalities) are not necessarily aligned. It would be useful to see results with a simple concatenation of embedding vectors as an alternative.

 - Line 240: The writing style could be improved, particularly in the phrase "modality 1 data x_1."

 mutually enhanced fusion - A major writing issue is that while the notation is presented for two modalities for clarity, it is not clear how the "mutually enhanced fusion" module would generalize when there are more than two modalities. Pairwise enhancement for an arbitrary number of modalities may be inefficient and needs further clarification.

 - Addition of Gaussian Noise (Line 286): It is unclear how adding Gaussian noise with zero mean and unit variance improves robustness. More explanation would help in understanding the rationale behind this choice. (no theoretical or experimental support or reference )

- Results in Figures 4 and 5: The results show that the performance is not sensitive to the number of experts. It would be insightful to see the distribution of weights assigned to each expert, to determine if the model is selectively using experts or simply averaging across them. Additionally, it is important to visualize that no single expert is consistently dominant across different scenarios.

**Questions:**

Please refer to weakness. Moreover, my main concern is the paper's limited contribution and novelty (considering the ICLR's quality). I will also read other reviewer's comments before making my final decision (my rating is for the current status of the paper).

---

> ### Comment · Reviewer_CgBN · 2024-11-26
> **reject**
>
> The authors do not respond, so I keep my score.

---

### Official Review · Reviewer_ooDY · 2024-11-04

**Soundness:** 2
**Presentation:** 3
**Contribution:** 2
**Rating:** 6
**Confidence:** 3

**Summary:**

The paper introduces Multimodal Open Set Recognition (MMOSR), extending the concept of Open Set Recognition (OSR) to multimodal data, which includes tasks that require processing multiple types of data simultaneously (e.g., image-text, audio-visual). The authors identify a key challenge in MMOSR: fusion degradation, where simply combining multimodal fusion methods with OSR leads to overly compact representations that hinder the model's ability to discriminate unknown samples.

**Strengths:**

(1)MMOSR fills a gap in handling multimodal data for open set recognition.
(2)The MRN effectively addresses fusion degradation through its dual fusion strategies.
(3)Comprehensive Evaluation: The paper provides thorough experimental validation across multiple datasets and scenarios, demonstrating the robustness and superiority of MRN.

**Weaknesses:**

(1) There is another works claiming the importance of the multimodal open-set learning.  Could the author talk about the difference between this work and the reference [1]?
(2) We can find some improvement comparing the visualizations from this work and existing works on the dataset Food-101 (Fig. 2 & Fig. 6). However, the performance in  Flower-102 seems not good.  Could the author show some results of visualizations of existing methods?
(3)  Could the author analyze the performance on CREMA-D which is not good.




[1] Towards Multimodal Open-Set Domain Generalization and Adaptation through Self-supervision

**Questions:**

See in weaknesses

---

> ### Comment · Reviewer_ooDY · 2024-11-27
> **Final Decision**
>
> Since the author did not provide any feedback, I decide to keep my score.

---

### Official Review · Reviewer_dSS5 · 2024-11-04

**Soundness:** 2
**Presentation:** 3
**Contribution:** 2
**Rating:** 5
**Confidence:** 4

**Summary:**

The paper tackles the challenge of OSR within multimodal settings. Traditional OSR models frequently encounter difficulties in accurately identifying unknown samples when multiple modalities are involved. To address this, the authors introduce a multimodal recognition network (MRN) aimed at enhancing OSR performance across diverse datasets. Experiments are used to verify the effectiveness of the proposed method.

**Strengths:**

(1). This paper introduces a new task, Multimodal Open Set Recognition (MMOSR), which is more practical compared to the traditional OSR setting.

(2). The paper identifies that combining traditional OSR methods with multimodal fusion leads to fusion degradation and designs a network capable of achieving comprehensive and informative representations.

**Weaknesses:**

(1). This paper illustrates the issue of fusion degradation after modality fusion using only visualization in Figure 2, lacking theoretical analysis and proof.

(2). The proposed MRN framework lacks novelty, as it merely integrates existing methods, i.e., Cross Attention and MOE, without substantial innovation.

(3). Compared to existing multimodal fusion methods (e.g. MLA), the improvements in various OSR task metrics presented in this paper are not sufficiently significant.

**Questions:**

(1). Could you elaborate further on the causes of the fusion degradation phenomenon and its relationship with the OSR task?

(2). Table 1 only compares the results of single models and simple fusion methods. Could you include a comparison with additional multimodal methods for further analysis?

(3). I recommend including a chapter index for each section in the appendix within the main text, such as detailed definitions of $g_I$ and $g_L$.

---

### Meta-Review · Area_Chair_7SPZ · 2024-12-17

**Metareview:**

The paper addresses Multimodal Open Set Recognition (MMOSR) and proposes the Multimodal Representation Reactivation Network (MRN) to mitigate fusion degradation, a challenge in combining multimodal data for OSR tasks. Strengths include introducing a practical MMOSR task and demonstrating experimental improvements on benchmark datasets. However, the weaknesses are significant: the method lacks technical novelty, relying on existing strategies like cross-attention and mixture of experts, without sufficient theoretical justification or insights into fusion degradation. Additionally, the experimental evaluation is incomplete, missing comparisons, ablations, and broader multimodal scenarios. Due to limited contributions and unconvincing results, I recommend rejecting the paper.

**Additional Comments On Reviewer Discussion:**

Since the authors did not prepare a rebuttal, no discussion was initiated.

---

### Decision · Program_Chairs · 2025-01-22

Reject